# Lightning at Jupiter pulsates with a similar rhythm as in-cloud lightning at Earth

Ivana Kolmašová [1,2] ✉, Ondřej Santolík [1,2], Masafumi Imai [3], William S. Kurth [4], George B. Hospodarsky[4], John E. P. Connerney[5], Scott J. Bolton[6] & Radek Lán[1]

Our knowledge about the fine structure of lightning processes at Jupiter was substantially limited by the time resolution of previous measurements. Recent observations of the Juno mission revealed electromagnetic signals of Jovian rapid whistlers at a cadence of a few lightning discharges per second, comparable to observations of return strokes at Earth. The duration of these discharges was below a few milliseconds and below one millisecond in the case of Jovian dispersed pulses, which were also discovered by Juno. However, it was still uncertain if Jovian lightning processes have the fine structure of steps corresponding to phenomena known from thunderstorms at Earth. Here we show results collected by the Juno Waves instrument during 5 years of measurements at 125-microsecond resolution. We identify radio pulses with typical time separations of one millisecond, which suggest step-like extensions of lightning channels and indicate that Jovian lightning initiation processes are similar to the initiation of intracloud lightning at Earth.

During the Jupiter flyby of Voyager 1 in 1979, the onboard radio receiver observed several seconds long radio wave signals at a slowly decreasing frequency from approximately 7 kHz down to 1 kHz. All these detections were obtained in the Io torus, and their similarity with the terrestrial lightning whistlers led to the discovery of Jovian lightning[1,2]. The cadence of these signals gave a maximum of one lightning per second, and the analysis of the whistler traces gave an upper limit of 40 ms for the time scales of underlying lightning processes. Optical instruments onboard Voyager 1 and subsequent spacecraft missions to Jupiter did not reach sufficient time resolution to resolve separate lightning phenomena. Their exposure times were too long to capture individual lightning strokes (Voyager 1, 2: 35–192 s[3,4], Galileo and Cassini: 6.4–179 s[5,6], New Horizons: 5 s[7]).

The lightning and radio emission detector onboard the Galileo Probe investigated characteristics of radio frequency signals (10 Hz to 100 kHz)[8]. The capability of the instrument to detect groups of impulsive signals was proven during its calibration against terrestrial lightning[9], showing the characteristics of both intracloud (IC) and cloud-to-ground (CG) discharges. During the descent phase, the probe recorded radio frequency signals from a distant source[10] which were dominated by low frequencies with long pulses of the order of a few hundred microseconds. It was found that the probe entered a spot with a dry and stable environment[11], and modeling results showed that there was a very low probability that the probe could have encountered local thunderstorms[12]. This implies that the signals received by the probe arrived from a distant storm located up to a few thousand kilometers from the descent trajectory[8].

The Juno spacecraft has orbited Jupiter since 2016, diving down to a few thousand kilometers above the clouds. Its Waves investigation instrument[13] discovered Jovian rapid whistlers[14] at frequencies below 20 kHz as a form of dispersed atmospherics at short time scales of less than a few milliseconds to a few tens of milliseconds, and Jupiter dispersed pulses[15] (JDP) below 150 kHz, propagating in the ordinary mode at even shorter time scales from less than 0.4 milliseconds to several milliseconds. Juno Waves data also allowed us to verify that the rapid whistlers are upward propagating signals, which have been shown to

[1]Department of Space Physics, Institute of Atmospheric Physics of the Czech Academy of Sciences, Prague, Czechia. [2]Faculty of Mathematics and Physics, Charles University, Prague, Czechia. [3]Department of Electrical Engineering and Information Science, National Institute of Technology (KOSEN), Niihama College, Niihama, Ehime, Japan. [4]Department of Physics and Astronomy, University of Iowa, Iowa City, Iowa, USA. [5]NASA/Goddard Spaceflight Center, Greenbelt, Maryland, USA. [6]Space Science Department, Southwest Research Institute, San Antonio, Texas, USA. ✉e-mail: iko@ufa.cas.cz

occur with an average cadence of up to a few whistlers per second, similar to observations from Earth-orbiting spacecraft above thunderstorms[14].

Moreover, the Stellar Reference Unit (SRU)[16] onboard Juno allowed the field of view of the camera to be panned at a rate of 1 pixel every 2.7 ms using 1 s exposures. This led to a recent observation of 5.4 ms long lightning, with inter-flash separation of tens of milliseconds[17]. The Microwave Radiometer (MWR)[18] onboard Juno showed that lighting sferics in the 600 MHz and 1.2 GHz frequency band have similar distribution over the surface of Jupiter as Jovian rapid whistlers and Jupiter dispersed pulses during the first nine close approaches to Jupiter[14]. MWR integrates each radiance measurement for 0.1 s[19], being not able to distinguish repetitive signals on milli-second scales.

In this work, we use nearly 5 years of data collected by the Juno Waves instrument and identify electromagnetic signals with a typical time separation of one millisecond and with a power law distribution of interpulse intervals. These time scales correspond to terrestrial in-cloud breakdown processes. The Juno dataset thus implies that Jovian lightning channels might extend in similar distinct steps after the lightning initiation, as observed during the initiation of intracloud lightning at Earth.

## Results

### Waves instrument onboard Juno spacecraft

The nominal part of the Juno mission[20] included 34 orbits. Every 53 days, Juno returned very close to Jupiter, and the perijove (PJ) altitude ranged from 3500 to 8000 km above the cloud tops. The first Juno close approach to Jupiter, during which scientific data were collected, occurred on 27 August 2016. The thirty-fourth close visit took place on 7 June 2021. The dataset we used for this analysis was collected by the Low Frequency Receiver (LFR) of the Juno Waves investigation instrument[13]. Its high-frequency part (LFR-hi) records 16.384 ms long electric field waveform snapshots with a sampling rate of 375 kHz. The low-frequency part (LFR-lo) records 122.88-ms-long electric and magnetic field waveform snapshots sampled at 50 kHz. The records are taken once per second during close approaches of Juno to Jupiter. The high- and low-frequency snapshots do not overlap, but the LFR-lo electric and magnetic snapshots do.

### Properties of pulse sequences

The entire dataset acquired below 5.5 Jovian radii during the nominal part of the Juno mission by the LFR-hi receiver (326,466 snapshots) was visually inspected for lightning-generated Jupiter dispersed pulses (JDPs) propagating in the free-space ordinary mode through the low-density regions[15] with number densities below 250 cm$^{-3}$, and the corresponding dataset from the LFR-lo receiver recorded up to 17th perijove (158,716 snapshots) was inspected for the presence of rapid whistlers[14] propagating in the whistler mode at frequencies below 20 kHz. In the LFR-hi data, we found 3182 snapshots with at least one JDP. Among them, we selected 375 snapshots with sequences of at least three JDPs, with well-distinguishable interpulse intervals, and with the same dispersion and frequency cutoff characteristic. This selection helps us to focus on radio waves, which propagate through the iono-spheric plasma to the spacecraft along the same path, and hence from the same source location of lightning processes. Three examples of LFR-hi snapshots with such groups of JDPs are shown in Fig. 1.

The shortest time separation of individual JDP traces that we were able to observe was about 170 μs. No distinct details of the underlying lightning processes at shorter time scales than these repetitive JDPs have been observed in the original waveforms down to the time scales of a few microseconds, defined by the data sampling interval. Source waveforms corresponding to examples in Fig. 1 and additional examples of JDP groups are shown in Supplementary Figs. 1 and 2, respectively. Note that although the dispersion characteristics and cutoffs

stay the same within each individual analyzed group of JDPs, they differ between the separate groups. This can be attributed to the different properties of radio wave propagation paths to Juno.

Time scales of Jovian lightning processes can be inferred from the time separation between individual electromagnetic pulses within each group. We determined the time delays between centers of neighboring JDP spectral traces whenever it was possible. The resulting dataset contains 2576 interpulse intervals. Similarly, we inspected the LFR-lo spectrograms in order to identify records containing groups of rapid whistlers[14].

We have found 4502 snapshots with at least one rapid whistler and 120 snapshots with groups of at least three whistlers exhibiting the same dispersion and upper cutoffs as well as distinguishable inter-whistler intervals. The examples of LFR-lo spectrograms with groups of whistlers are shown in Fig. 2. We estimated the time delay between centers of neighboring clear whistler traces and obtained a set of 482 inter-whistler intervals. The shortest distinguishable time separation of individual whistler traces was about 1.3 ms. Examples of whistler traces, which were intense and narrow enough to be included in the estimation of inter-whistler intervals, are shown in Fig. 2. The selection process leading to the final dataset is, for clarity, also shown in Table 1. A map of detections of repetitive lightning signals, obtained using vertical straight-line projections of Juno position down to the 1-bar level for JDPs and as projections along the magnetic field for whistlers, are displayed in Fig. 3a by black and blue open circles for groups of JDPs and rapid whistlers, respectively. For comparison, orange '+' signs show previously published projections of Juno's position for LFR-lo snapshots containing at least one rapid whistler during PJ1-8[14]. It is clear that the lightning processes, which emitted multiple JDPs or groups of rapid whistlers, were located at middle and higher latitudes, similar to detections of all rapid whistlers reported from the first quarter of the nominal part of the Juno mission[14]. However, while groups of rapid whistlers were observed closer to the planet than about 110,000 km, the groups of JDPs occurred in snapshots also acquired at larger distances from Jupiter, up to 260,000 km (Fig. 3b).

The number of JDPs in individual 16.384 ms long snapshots (Fig. 3c) reached the maximum value of 25. The group of five pulses was the most frequent one, which is above the lower limit of three pulses imposed by our analysis method. This is different for rapid whistlers, which occur most often at this limit of three whistlers within the 122.88-ms snapshots, and the fraction of their higher numbers then approximately exponentially decreases for up to 13 whistlers per snapshot.

### Distribution of interpulse intervals

The time separation between signal traces estimated within all JDP groups varied from 0.17 to 11.72 ms with a mean value of 1.37 ± 1.27 ms and a median value of 0.95 ms. The distribution of all interpulse intervals is displayed in Fig. 3d. It is clear that the probability density must artificially decrease at delays above 5 ms because of the limitations imposed by the finite length of the snapshot. The probability density function (PDF) is therefore completed by results obtained from the analysis of groups of rapid whistlers.

The time separation between rapid whistler traces estimated within all whistler groups varied from 1.8 to 94 ms. The probability density function of obtained values approximately follows a power law PDF($\delta$) = $A\ \delta^B$, where $\delta$ is the interpulse interval and the power law exponent $B = -1.87 ± 0.07$ for $\delta$ above 4 ms. Careful normalization of the composed probability density function based on both the JDP and whistler data (see "Methods"−"Calculation of the probability density function") allows us to fit the power law exponent for $\delta$ above 1 ms, obtaining an exponent $B$ of $-1.89 ± 0.03$, while using only the JDP data for $\delta$ between 1 and 4 ms, the power law exponent $B$ is close to $-1.85$ with a standard deviation of 0.08. Note that these results are very consistent and clearly exclude a random distribution of independent

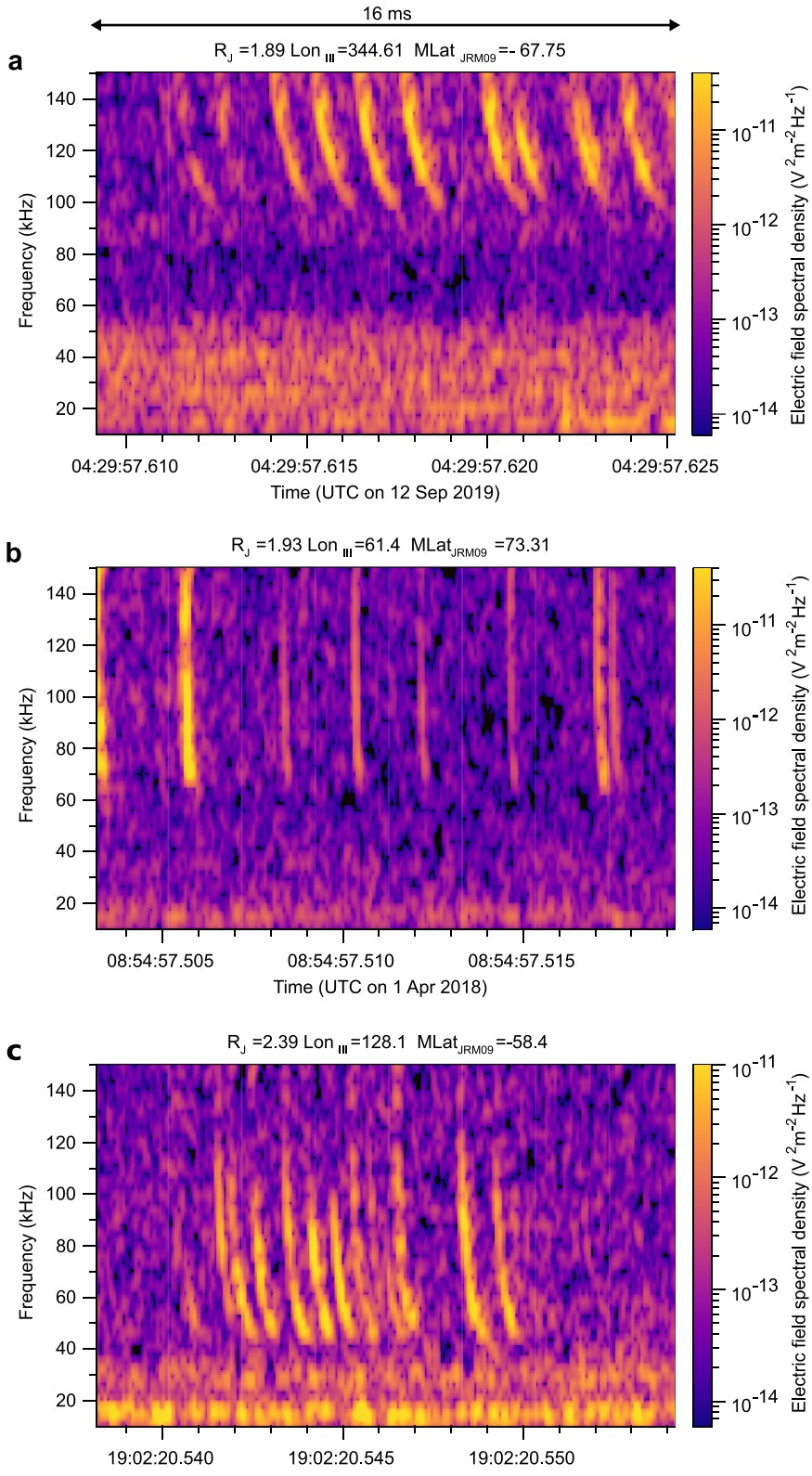

**Fig. 1 | Examples of groups of JDPs.** Frequency–time power spectrograms of the electric field fluctuations. **a** Snapshot recorded on 12 September 2017 after 04:29:57 UTC (Coordinated Universal Time) at a distance of 1.89 $R_J$ (Jovian radii) contains 10 JDPs. **b** Snapshot recorded on 1 April 2018 after 08:54:57 UTC at a distance of 1.93 $R_J$ contains 8 JDPs. **c** Snapshot recorded on 17 February 2020 after 19:22:20 UTC at a distance of 2.39 $R_J$ contains 13 JDPs.

events, where the probability density of time delays would be decreasing exponentially. This is, however, valid only for time scales below 100 ms, while the exponential distribution was previously found[4] at large time scales well above 100 ms.

The delays below 100 ms therefore reflect more complex and interlinked lightning phenomena than just a random occurrence. One of the distinct properties of our dataset is the similarity of pulses occurring within each separate snapshot. We therefore tested

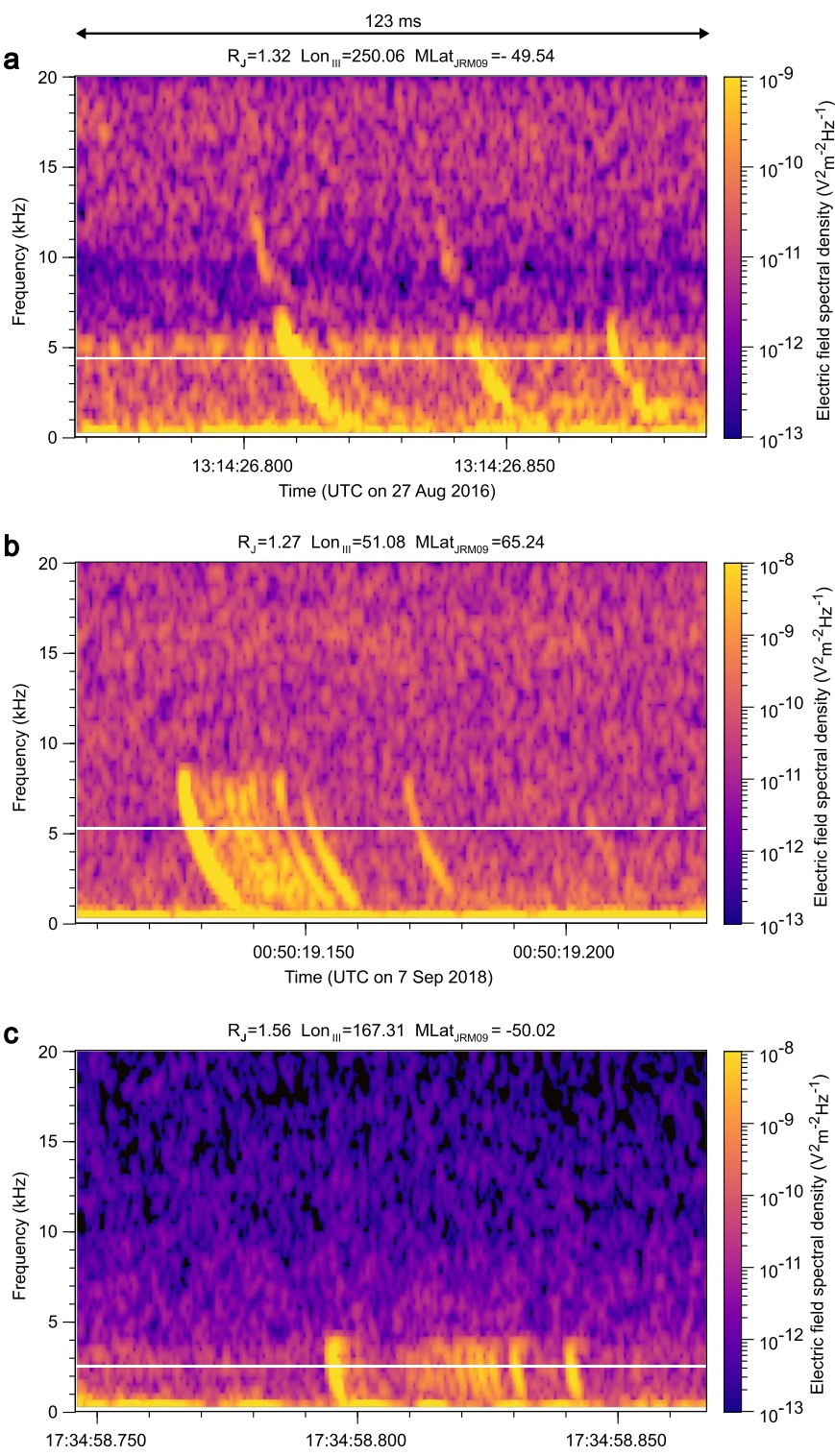

**Fig. 2 | Examples of groups of whistlers.** Frequency–time power spectrograms of the electric field fluctuations. **a** Snapshot recorded on 27 August 2016 after 13:14:26 UTC at a distance of 1.32 $R_J$ contains three whistlers. **b** Snapshot recorded on 7 September 2018 after 00:50:19 UTC at a distance of 1.27 $R_J$ contains seven whistlers. **c** Snapshot recorded on 21 December 2018 after 17:34:58 UTC at a distance of 1.56 $R_J$ contains 10 whistlers. White lines indicate the local proton cyclotron frequency.

the consistency of obtained results by calculating the average delays for each individual group of pulses separately and thus avoiding possible bias linked to the presence of long pulse trains. These average values (Supplementary Fig. 3a) varied from 0.43 to 6.66 ms with a median of 1.49 ms, which is consistent with results from all delays.

Another distinct property of the observed groups of pulses is nearly regular interpulse intervals within some of them, as are most of those in Fig. 1. One-fourth of groups (90 out of 374) fulfilled the criterion that the standard deviation of their interpulse intervals is smaller than one-half of their mean value. Within this subset of regular pulse trains, we found 643 interpulse intervals varying from 0.20 to

**Table 1 | Data selection: overview of the selection process, which resulted in the final dataset of sequences of at least three JDPs or three whistlers**

| Juno Waves receiver | Number of available snapshots | Number of snapshots with at least one JDP/whistler | Number of snapshots with at least three JDPs/whistlers | Number of inter-pulse intervals | The shortest distinguishable interpulse interval |
|---|---|---|---|---|---|
| LFR-Hi | 326,466 | 3182 | 375 | 2576 | 170 µs |
| LFR-Lo | 158,716 | 4502 | 120 | 482 | 1.3 ms |

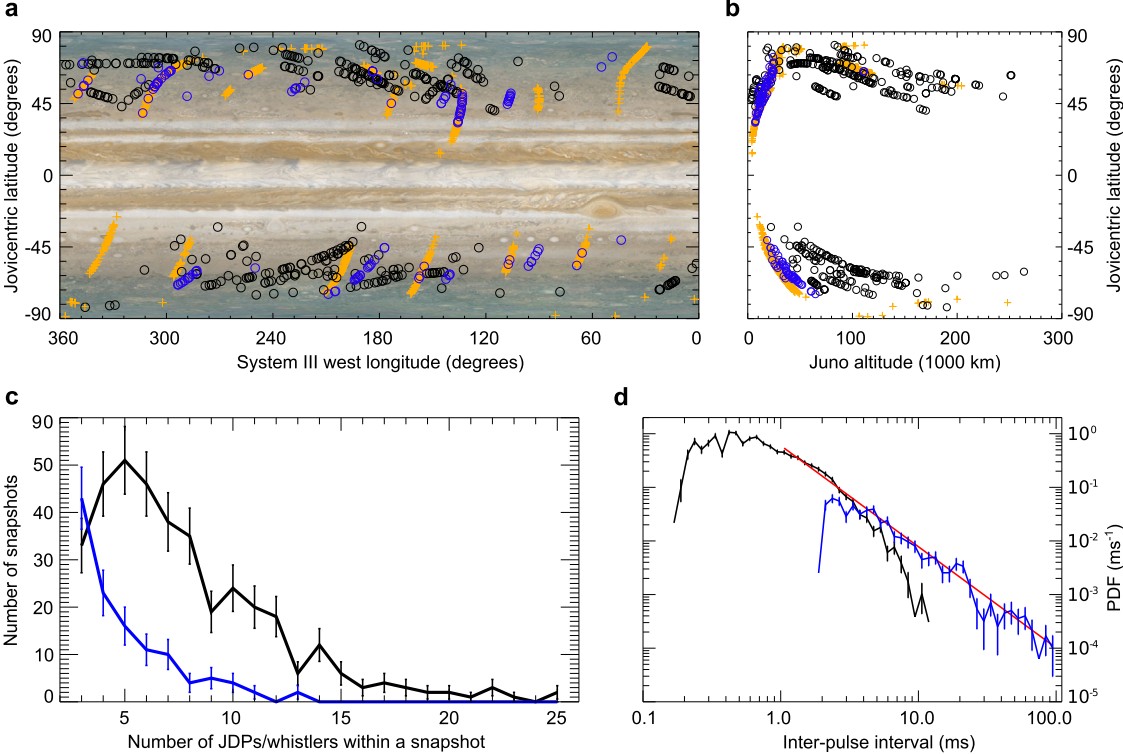

**Fig. 3 | Distribution of interpulse intervals. a** Map of detections of groups of JDPs as vertical projections of Juno positions on the 1 bar level (black circles) and groups of rapid whistlers (blue circles) as projections along the magnetic field[45]. Orange crosses display previously published positions of rapid whistlers during PJ1-8[14]. **b** Altitudes of Juno when detecting multiple JDPs (black circles) or multiple whistlers (blue circles) as a function of the Jovicentric latitude of the corresponding projections from panel (**a**). **c** Distribution of the number of pulses or whistlers in snapshots containing multiple JDPs (black) and multiple whistlers (blue). The error bars correspond to the standard deviations of the expected Poisson distribution for counts of JDPs/whistlers. **d** Probability density function (PDF) of JDP interpulse intervals (black) and separation between whistler traces (blue). Error bars correspond to the standard deviations of the Poisson statistics of the underlying counts of occurrences (see "Methods"−"Calculation of PDF"). The solid red line shows a power law fit for delays above 1 ms and below 4 ms on JDP data and for delays above 4 ms on rapid whistlers. The Jovian image in panel (**a**) is available at https://www.planetary.org/space-images/merged-cassini-and-juno. The source data are provided as a Source Data file. The relevant codes are available at https://data.mendeley.com/datasets/s2rg9ddb24.

5.22 ms with an average value of 1.28 ± 0.80 ms and a median value of 1.10 ms (Supplementary Fig. 3b), which is again consistent with results obtained for the entire dataset.

## Discussion

Our results show that typical time scales of repetitive radio wave signals of Jovian lightning are around 1 ms. In about one-fourth of observed cases, we capture regular sequences of pulses, similar to stepping processes at Earth. Globally, the probability density of longer interpulse delays decreases according to a power law with an exponent of −1.9 up to delays of at least 100 ms. The characteristics of the other Juno instruments (MWR and SRU), which are also observing Jovian lightning, do not allow for revealing any repetitive processes at the millisecond time scales because of lower instrumental resolution or longer exposure times.

The signals recorded by the Galileo Probe might be compatible with our observations, but the uniqueness of the Galileo Probe measurements does not allow us to compare the results more thoroughly.

Due to a long propagation below the ionosphere, the high-frequency portion of the signal was probably already attenuated.

Looking further into analogies with measurements of signals emitted by terrestrial lightning phenomena, we must consider the time scales of electromagnetic signals generated during different evolution stages of terrestrial lightning flashes. A comparison of signals generated by Jovian lightning with those observed by Earth-orbiting spacecraft would be necessarily influenced by the differences in propagation to the spacecraft through ionospheres with different properties on both planets and the time scales of their causative lightning phenomena can be analyzed in detail only in broadband recordings of ground-based radio receivers.

These time scales vary from units of microseconds (width of an individual dart-stepped leader pulse) to several hundred milliseconds (duration of multi-stroke lightning flashes)[21]. An analogy with CG lightning might be disputable for the Jovian atmosphere with no solid surface, but for completeness, we consider time scales of all known sequences of impulsive signals:

a.  groups of strokes in multi-stroke CG lightning flashes, where strokes are usually separated by several tens of milliseconds (globally $55 \pm 2$ ms for subsequent strokes flowing in pre-existing channels and $65 \pm 4$ ms for subsequent strokes with a new ground contact[22]);

b.  trains of pulses accompanying the initiation of normal IC flashes, where the pulses are separated by several hundreds of microseconds up to a few milliseconds;[23,24] the average propagation speed was $4 \times 10^5$ m/s;[25]

c.  trains of pulses appearing during initiation of attempted leaders (also called inverted IC flashes) or negative CG lightning, where pulses are separated by several tens to a few hundred of microseconds[24,26] (73 µs and 152 µs long average interpulse intervals were respectively found in the electromagnetic data from Florida[27] and Malaysia[28]);

d.  sequences of stepped leader pulses of CG lightning with inter-pulse separation of several units to a few tens of microseconds[29] ($0.2$–$15.7$ µs with an average value of $3.3$ µs corresponding to many simultaneously active leader branches in Florida, US[30], or an average inter-step interval of $16.4$ µs identified within optical detections of individual branches in Florida, US[31], and $13.9$–$23.9$ µs with a mean value of $17.4$ µs in electromagnetic measurements of individual branches in China[32]);

e.  pulse sequences of stepping recoil leaders or dart-stepped leaders with an interpulse separation of units of microseconds ($6.1 \pm 3.1$ µs and $5.1 \pm 1.8$ µs, respectively in Florida and Arizona, US[33], typically $6$–$8$ µs as derived for the lightning standard purposes[34]; average propagation speeds of $1.4$–$2.2 \times 10^6$ m/s and $6.4 \times 10^6$ m/s were reported for downward and upward leader propagation[35]).

Even though there are, on average, three to six strokes in terrestrial multi-stroke CG flashes[36] and the groups of JDPs predominantly consisted of five pulses, the Jovian pulse groups were very unlikely generated by multi-stroke lightning flashes. In such case, the Jovian strokes would appear in about 30 times faster succession than on the Earth[29]. This scenario seems to be therefore improbable. It is also in contradiction with modeling results which predicted that a time interval of 10 s was needed to establish conditions for initiation of a subsequent discharge in Jovian water clouds[37].

Only comparison with terrestrial IC processes might therefore be relevant to Juno observations. The initial breakdown pulses are believed to appear during the initiation phase of the majority of both CG and IC terrestrial lightning flashes[38]. The spectrogram in Fig. 4a shows an example of a sequence of initial breakdown pulses belonging to an inverted IC flash recorded by a broadband magnetic field antenna (5 kHz to 90 MHz) installed at the Milešovka observatory in Czechia[39] (known for frequent lightning, as it is reflected in the German version of its name "Donnersberg"–"thunder mountain"). Figure 4b shows a spectrogram of the initiation of a normal IC flash recorded by a similar antenna system installed at the nearby Dlouhá Louka observatory in Czechia. Magnetic field waveforms corresponding to spectrograms in Fig. 4, together with a waveform showing a multi-stroke negative CG flash, are plotted in Supplementary Fig. 4. The time separations of observed pulses in examples shown in Fig. 4 correspond to the typical time scales mentioned in the list of repetitive lightning phenomena (points c and b).

Previous studies[37,40] placed initiation of Jovian lightning inside clouds composed of water and ice and located at a temperature range between 250 and 270 K. These studies also showed that a strong convective motion driven by the internal heat is favorable for separation of charged water and ice particles and for an establishment of electric fields sufficient for lightning initiation. The breakdown fields calculated for Jovian water clouds located at different altitudes ($6 \times 10^5$ V/m at 1 bar level to $2.3 \times 10^6$ V/m at 5 bar level[37]) are very close to breakdown fields observed around or below 1 bar level in terrestrial thunderclouds ($3 \times 10^5$–$1 \times 10^6$ V/m). We can therefore assume that the initiation processes, including the average leader velocities at Jupiter and Earth, might be similar.

Based on the similarity of time scales of the observed groups of JDPs or Jovian whistlers with these terrestrial initial breakdown pulse trains, a probable scenario could be offered: The Jovian pulse sequences might be emitted during step-like extensions of Jovian lightning channels after the lightning initiation, similar to initiation processes of lightning at Earth. If we assume that the evolving discharges propagate in the Jovian water clouds at a similar average velocity of $10^5$–$10^6$ m/s as the terrestrial intracloud lightning leaders[21], the average prolongation of Jovian lightning channels might happen in steps of several hundreds to a few thousand meters long (see "Methods"–"Estimation of the lengths of the steps"). In contrast with the sferics detected by the Galileo Probe, our dataset does not indicate a substantial charge transfer. This is not surprising, as a smaller amount of charge is expected to be transferred during the stepping process. Note that the velocity of individual steps might be–similarly to on Earth–much faster than the average velocity of the leader prolongation, which also includes quiet stages between the steps[41]. Our results indicate that Jovian lightning is initiated at a larger spatial scale compared to processes preceding cloud-to-ground lightning in the Earth's atmosphere, but the process might be comparable to the initiation of normal intracloud lightning.

## Methods

### Power spectrograms

Power spectrograms from Juno measurements in Figs. 1 and 2 were obtained from the original electric (all LFR-hi snapshots and 95 LFR-lo snapshots) and magnetic field (25 LFR-lo snapshots) waveform data acquired by Juno using the Fast Fourier Transform (FFT) based on 128 waveform samples with a von Hann window, shifted by 32 samples in every time step of the spectrogram, giving 75% overlapping rate. The frequency and time steps are then, respectively, 2.93 kHz and 85 µs for JDPs in Fig. 1, with a nominal time resolution (full width at half maximum of the windowed signal power) of 125 µs. For rapid whistlers in Fig. 2, the frequency and time steps respectively are 391 Hz and 640 µs, with a nominal time resolution of 940 µs.

### Calculation of the probability density function (PDF)

The PDF values of delays between neighboring pulses in Fig. 3d are calculated from counts of occurrences $N_{Ji}$ and $N_{Wi}$ for JDPs and rapid whistlers, respectively, in each bin $i$ of delays:

$$f_{Ji} = \frac{N_{Ji}}{\delta_i M_J \tau_J} \frac{1}{L} \tag{1}$$

$$f_{Wi} = \frac{N_{Wi}}{\delta_i M_W \tau_W K_W} \frac{1}{L} \tag{2}$$

where $f_{Ji}$ and $f_{Wi}$ respectively are probability density values for JDPs and whistlers. $\delta_i$ is the width in milliseconds of each bin of delays, $M_J$ is the total number of analyzed waveform snapshots recorded below 5.5 Jovian radii by the LFR-hi receiver, $\tau_J$ is the duration of each snapshot. The total duration $M_J \tau_J = 534$ s of waveform captures at radial distances below 5.5 Jovian radii was used for our analysis of JDPs. Similar values $M_W$ and $\tau_W$ for the LFR-lo receiver give the total duration $M_W \tau_W = 19503$ s of waveform captures included in the analysis of rapid whistlers. The normalization coefficient $L$ is set so that the integral of the probability density function is normalized to unity:

$$L = \sum_{i=1}^{i_4} \frac{N_{Ji}}{M_J \tau_J} + \sum_{i=i_4+1}^{i_T} \frac{N_{Wi}}{M_W \tau_W K_W} \tag{3}$$

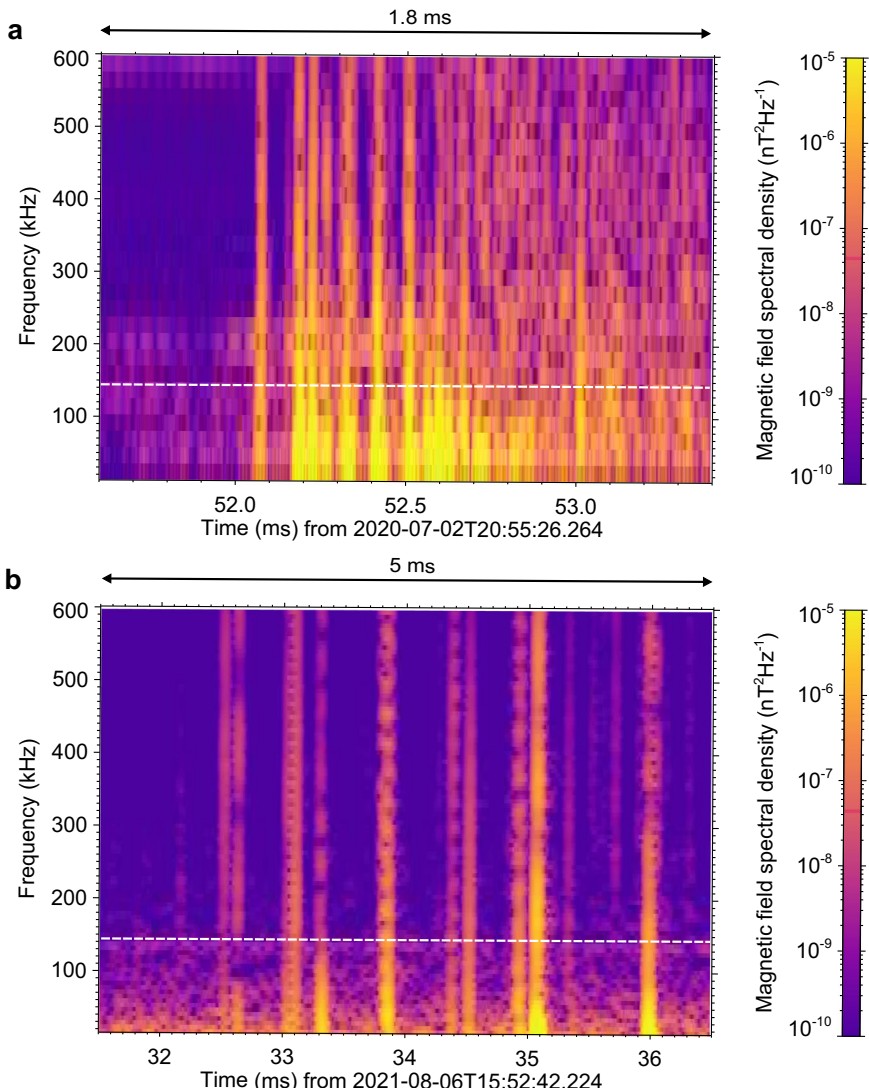

**Fig. 4 | Terrestrial ground-based broadband measurements. a** The frequency–time power spectrogram of magnetic field fluctuations showing the 1.8 ms long detail of the initiation of an inverted IC flash occurring on 2 July 2020 at 20:55:26 UTC. **b** The frequency–time spectrogram of magnetic field fluctuations showing the 5 ms long detail of the initiation of an IC flash occurring on 6 August 2021 at 15:52:42 UTC. In comparing with JDPs of Fig. 1 in the spectral scale, white dashed lines indicate the upper frequency limit of the Waves LFR-Hi measurements. The source data are available at https://data.mendeley.com/datasets/w629b9nyx2[43].

where $i_4$ denotes the index of the bin containing the delay of 4 ms, and $i_T$ is the total number of bins. The physical meaning of the coefficient $L$ is the average cadence of detected lightning pulses per second, obtained as $L = 0.47\,\text{s}^{-1}$ from our dataset. Finally, the calibration coefficient $K_W$ is introduced to reflect the lower detection ability for rapid whistlers linked to the higher natural background in the LFR-lo band and also to a different altitude distribution of rapid whistlers in comparison to JDPs, which are better distributed over the analyzed range of altitudes. A value of $K_W = 0.21$ was estimated from the requirement of continuity of power law fits to $f_{Ji}$ and $f_{Wi}$ values.

### Terrestrial measurements

Examples of measurements of terrestrial lightning processes in Fig. 4 were obtained from measurements of Shielded Loop Antenna with Versatile Integrated Amplifier (SLAVIA) devices, yielding broadband magnetic field waveforms sampled at 200 MHz, covering a frequency range between 5 kHz and 90 MHz. Power spectrograms of SLAVIA data were again obtained using FFT, but this time based on 32768 waveform samples with a von Hann window, resulting in a frequency step of 6.1 kHz and a nominal time resolution of 60 μs.

### Estimation of the lengths of the steps

Even if the rapid extension of the leader in individual steps might be as fast (~$10^8$ m/s) on Jupiter as on Earth[41], we are not able to confirm this velocity using the existing Juno data. The ionospheric dispersion does not allow us to estimate the duration of rising edges of individual pulses with a sufficient time resolution. That is why we limit our estimation to an average speed of the leader prolongation happening over a group of leader steps, i.e., including the intervals between the pulses when we assume that the leaders do not extend. We further assume that the average speed of the Jovian leader prolongation is similar to the average speed of leaders propagating in the terrestrial water clouds at a similar pressure. If we combine this average velocity of $10^5$–$10^6$ m/s with the interpulse intervals resulting from our study (typically a few milliseconds, Fig. 3d), we obtain the average step length of several hundreds to a few thousand meters.

### Data availability

The Juno Waves calibrated burst waveform full-resolution dataset includes all high-rate science waveform information calibrated in units of electric or magnetic field for the entire Juno mission and has been

deposited at https://doi.org/10.17189/1522461[42]. The Jovian image in Fig. 3a is freely available at https://www.planetary.org/space-images/merged-cassini-and-juno (provided by NASA/JPL-Caltech/SSI/SwRI/MSSS/ASI/INAF/JIRAM/Björn Jónsson). The source data for Fig. 3 is provided with this paper in the Excel sheets in the Source Data file. The ground-based data displayed in Fig. 4 are available at https://data.mendeley.com/datasets/w629b9nyx2[43], doi:10.17632/w629b9nyx2.1. The data generated in this work are available in the Source data zip file (inter-whistler intervals.csv and inter-JDP intervals.csv). Source data are provided with this paper.

## Code availability

The custom codes were written in IDL® (a product of Exelis Visual Information Solutions, Inc., a subsidiary of Harris Corporation), were used for generating figures, and are not central to the paper. The codes are available at https://data.mendeley.com/datasets/s2rg9ddb24[44], https://doi.org/10.17632/s2rg9ddb24.1.

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

## Acknowledgements
The authors would like to acknowledge all members of the Juno mission team, especially the engineers and staff of the Juno Waves instrument. The research of W.S.K. and G.B.H. was supported by NASA through Contract 699041X with the Southwest Research Institute. We acknowledge the use of the Space Physics Data Repository at the University of Iowa, supported by the Roy J. Carver Charitable Trust. The work of I.K. and O.S. was supported by the LTAUSA17070 and GACR 20-09671S grants. The work of R.L. was supported by the European Regional Development Fund-Project CRREAT (CZ.02.1.01/0.0/0.0/15_003/0000481). The authors thank Gerhard Diendorfer for the EUCLID lightning data.

## Author contributions
I.K., O.S., and W.S.K. designed the study. The manuscript was written by I.K. and O.S. with input from all authors. I.K. and M.I. independently performed the extensive search for Jovian rapid whistlers and Jupiter dispersed pulses in the Waves burst dataset and converged to a common list of events. W.S.K. and G.B.H. provided consultations on data analysis. W.S.K. is responsible for the Juno Waves instrument. J.E.P.C. provided the planetary magnetic field measurements. S.J.B. is the Principal Investigator of the Juno spacecraft. R.L. is responsible for broadband ground-based measurements and their data storage.

## Competing interests
The authors declare no competing interests.
