## [Peer Review File · Nature Communications]

REVIEWER COMMENTS

Reviewer #1 (Remarks to the Author):

Review of "Lightning at Jupiter pulsates with a similar rhythm as in-cloud lightning at Earth"

This work takes a statistical view of the recent NASA Juno prime mission radio wave data looking at the timing of the electromagnetic pulses produced by lightning. In this case, these pulses are observed as Jupiter dispersed pulses and rapid whistlers. The authors show that the pulse timing is similar to that at Earth for intracloud lightning and therefore that the lightning at Jupiter could follow a similar morphology giving us a significant insight into how the lightning process works on a Gas Giant which has potential applications within our Solar system and beyond.

The work shown broadly supports the authors conclusion that Jovian lightning could be similar to intracloud processes at the Earth.

However, the discussion of the comparison to Earth measurements needs to include more discussion of the timing statistics in the referenced works [20,21,22,23] to justify the conclusions properly. How tightly bound are the timings quoted in the list starting at L206 for example? Ideally a similar analysis to that performed in the current work would show the proposed similarities best.

I would also like to see a short discussion on what the differences in the atmosphere/clouds are thought to be at Jupiter that might affect the timing (or not as the case may be) of the lightning.

The methods used are appropriate and the methods section appears to be sufficiently detailed.

I have a few minor comments:

Although there is a direct correlation between the inter pulse timing of these signatures observed in space and those observed on the ground - why do the comparison to ground measurements rather than to space at Earth? Is it possible to compare DPs and whistlers more directly than just the timings of electromagnetic bursts?

Typographical errors: A small number of errors, see attached file

Reviewer #2 (Remarks to the Author):

"Lightning at Jupiter pulsates with a similar rhythm as in-cloud lightning at Earth"

by Kolmašová et al.

Review by Ulyana Dyudina

The paper presents Juno observations on electromagnetic signals from lightning on Jupiter. From the frequency and cadence of the signals the authors argue about similarities of the lightning stepped structure with the intracloud lightning on the Earth.

The paper gives an interesting update to the knowledge about lightning on Jupiter.

I believe the paper needs substantial clarifications.

I provide comments as yellow sticky notes on the attached manuscript.

In addition here are some suggestions that would make the paper easier to read for non-specialists:

There is quite a bit of data values quoted right in the paper text. It would be helpful to give a reader more systematic view on the data using tables, equations and figure details.

The comparison with just one data set on terrestrial lightning is interesting. However it would be important to know how other published observations on terrestrial lightning agree with the observations presented in this paper.

Reported stepped structure of jovian lightning and its timing is a robust and convincing result.

However, the main conclusion of the paper is about the length of lightning steps.

The assumption of similar speeds for lightning used for the the steps' length calculation needs to be better justified and discussed in the paper text.

It would be very interesting to discuss the range of possible lightning propagation speeds and corresponding step lengths.

All the best!

Ulyana Dyudina

Reviewer #3 (Remarks to the Author):

Kolmasova et al. (2023), Lightning at Jupiter pulsates with a similar rhythm as in-cloud lightning at Earth.

I am very impressed by this work. Electric field waveforms produced by Jovian lightning (Sup. Fig. 1), as well as the corresponding spectrograms (Fig. 1) and the associated analysis, certainly deserve to be published in Nature Communications. My comments aimed at clarification of several points are found below.

1. I think the authors could give more credit to the pioneering Galileo probe experiment. As far as I know, in that experiment, the first wideband (100 Hz to 100 kHz) field signatures of Jovian lightning were obtained below the ionosphere with an instrument (LRD) that was calibrated against terrestrial lightning (see Lanzerotti et al., 1992, 1996). Yes, their signals were dominated by the lower-frequency components, in contrast with this study, in which the signals do not appear to indicate any significant charge transfer. Maybe the reason for disparity is the ionosphere that did not allow the lower-frequency components to be recorded in the present study. In any event, this is worth discussing in the paper, particularly the lack of detectable charge transfer in the present study and the fact that LRD was fully capable of recording impulsive signals (see Fig. 11 of Lanzerotti et al., 1992), but did not see any.

References

Lanzerotti, L.J., Rinnert, K., Dehmel, G., Gliem, F.O., Krider, E.P., Uman, M.A., and Bach, J. 1996. Radio frequency signals in Jupiter's atmosphere. *Science*, 272: 858-60.

Lanzerotti, L.J., Rinnert, K., Dehmel, G., Gliem, F.O., Krider, E.P., Uman, M.A., Umlauf, G., and Bach, J. 1992. The lightning and radio emission detector (LRD) instrument. *Space Sci. Rev.*, 60: 91-109.

2. Lines 243-248 (see also Lines 47-49): Some clarification is needed here. First, the initial breakdown (IB) in terrestrial lightning is usually hidden inside the cloud (not imaged by high-speed cameras), and there are different theories on the nature of this process (including some kind of runaway breakdown process). In view of the relatively poor understanding of the IB on Earth, is it prudent to state that IB processes on Jupiter and on Earth are similar? Further and more importantly, I do not see how the average leader extension speed (10^5 to 10^6 m/s) and the average inter-pulse interval (about 1 ms) can be used for estimating the step length. It is usually assumed (see, for example, Carl Baum's work) that the stepped-leader channel extends only (or primarily) during the step-formation process, as opposed to inter-step intervals. The step-formation process is much faster (10^7 to 10^8 m/s) than the overall leader extension. Please clarify (maybe in the Methods section).

3. Lines 217-218: Actually, the average number of strokes per flash is 3 to 5. The quoted range from 2 to 3 is based on ALDIS data, and all LLSs underestimate this parameter. I found a recent summary of non-LLS measurements in Table 4.1 of Rakov (2016).

Reference

Rakov, V.A. 2016. *Fundamentals of Lightning*, Cambridge.

4. Lines 213-214: The average time interval between stepped-leader steps is usually a few tens of microseconds (see, for example, Table 7.1 of Rakov (2016)), not "units of microseconds".

In summary, I recommend this paper be published after revision to address the above comments.

We thank Ulyana Dyudina and two other reviewers for careful reading of the manuscript and for their helpful comments and suggestions. We responded to all of them and revised the manuscript accordingly. (The lines are related to the manuscript with tracked changes.)

Responses to Reviewers (in blue) with Reviewers' comments in black.

=====

Reviewer #1

This work takes a statistical view of the recent NASA Juno prime mission radio wave data looking at the timing of the electromagnetic pulses produced by lightning. In this case, these pulses are observed as Jupiter dispersed pulses and rapid whistlers. The authors show that the pulse timing is similar to that at Earth for intracloud lightning and therefore that the lightning at Jupiter could follow a similar morphology giving us a significant insight into how the lightning process works on a Gas Giant which has potential applications within our Solar system and beyond. The work shown broadly supports the authors conclusion that Jovian lightning could be similar to intracloud processes at the Earth.

However, the discussion of the comparison to Earth measurements needs to include more discussion of the timing statistics in the referenced works [20,21,22,23] to justify the conclusions properly. How tightly bound are the timings quoted in the list starting at L206 for example? Ideally a similar analysis to that performed in the current work would show the proposed similarities best. The time scales of different lightning phenomena listed in the overview of terrestrial lightning processes generating electromagnetic pulse trains have been statistically analysed in the past. The values obtained at different locations and seasons can slightly differ. We added more statistical information about these time scales on lines 236-254 together with relevant references.

I would also like to see a short discussion on what the differences in the atmosphere/clouds are thought to be at Jupiter that might affect the timing (or not as the case may be) of the lightning. We do not expect very different electrification properties and especially the range of average leader velocities on Jupiter and at the Earth taking into account similar conditions for the leader initiation and similar breakdown electric fields (ref 39 - Yair et al., 1995). We added a new paragraph in the discussion section on lines 287-295:

The methods used are appropriate and the methods section appears to be sufficiently detailed.

I have a few minor comments:

Although there is a direct correlation between the inter pulse timing of these signatures observed in space and those observed on the ground - why do the comparison to ground measurements rather than to space at Earth? Is it possible to compare DPs and whistlers more directly than just the timings of electromagnetic bursts?

This comparison is possible but it would add more components in the system, namely the properties of the wave propagation through the relatively dense terrestrial ionosphere. In this paper, we tried to concentrate on the timing of the underlying discharge processes in the atmosphere of Jupiter but we have to accept here that the investigated electromagnetic signals propagate through the Jovian ionospheric plasma to the spacecraft. These propagation effects can be very different compared to Earth, as the magnetic field at Jupiter is much larger and the plasma density lower. Therefore, it makes sense to avoid at least the propagation effects induced by the terrestrial ionosphere and use the ground based measurements for our purpose. We added an explanation at the beginning of the comparison section on lines 223-229.

Typographical errors:

90 Time scales of Jovian lightning processes can be inferred **from** the time separation between
Corrected on line 90

94 . These lightning generated signals last for **from**
95 of several milliseconds to several tens of milliseconds in ~~the~~ contrast to several seconds long
Corrected on line 95

99 second, similar to observations by at Earth **from** orbiting spacecraft above thunderstorms
Corrected on line 99

=====

Reviewer #2

Review by Ulyana Dyudina

The paper presents Juno observations on electromagnetic signals from lightning on Jupiter. From the frequency and cadence of the signals the authors argue about similarities of the lightning stepped structure with the intracloud lightning on the Earth. The paper gives an interesting update to the knowledge about lightning on Jupiter. I believe the paper needs substantial clarifications. I provide comments as yellow sticky notes on the attached manuscript.

Lines 46-49: I would soften this conclusion by switching the order: "If the lightning channels' prolongation speed..., then they can extend..."

We rather changed the abstract according to the reviewer's suggestion by stressing the convincing conclusion and we moved estimation of the step lengths to the discussion. The last sentence of the abstract now reads as follows:

"The Juno data thus suggests similar step-like extensions of Jovian lightning channels after the lightning initiation, as observed during initiation processes of intracloud lightning at Earth.

Lines 56-64: There are multiple long numbers in the text that are hard to follow for non-specialist. I suggest putting this in a table or figure.

We have collected all the numerical values linked to the selection process of the final dataset in Table 1.

Line 88: It is not clear what features in fig 1 correspond to low densities.

The presence of low-density regions was revealed by the previously published analysis of JDPs (ref. 3 – Imai et al. 2019) and it is not directly attributable to any features in Fig. 1. In order to avoid confusion, we now moved this piece of information to the introduction section, line 64 "... Jupiter dispersed pulses (JDPs) propagating in the free-space ordinary mode through the low density regions with the number densities below 250 cm^{-3} ."

Line 90: inferred from
Corrected on line 90.

Line 94: It is not clear for non-specialist how whistlers are different from JDP. Give a one- sentence introduction.

We added an information about the differences in the propagation modes of JDPs and whistlers. The sentence now reads as follows on lines 64-67:

“...was visually inspected for lightning generated Jupiter dispersed pulses (JDPs) propagating in the free-space ordinary mode through the low density regions, and the corresponding data set from the LFR-lo receiver was inspected for the presence of rapid whistlers propagating in the whistler mode at frequencies below 20 kHz.”

Line 102: What is an explanation for sferics being observed less often? Why are sferics important for this paper?

The MWR sferics are observed less often because MWR clearly has much lower sensitivity for detection of lightning signals than Waves. It is linked to the frequency contents of the lightning radio signals which are generally less intense at MWR frequencies with respect to the sensitivity of the MWR sensor. For the purpose of this study, MWR data are unusable because of a long integration time of the instrument. We moved the sentence about MWR to the discussion section on lines 203-206. Now it reads as follows:

“Comparison with the Microwave Radiometer (MWR) onboard Juno shows that lightning sferics in the 600 MHz and 1.2 GHz frequency band have similar distribution over the surface of Jupiter. However, MWR integrates each radiance measurement for 0.1 s being not able to distinguish repetitive signals on millisecond scales.”

Line 121: Are the vertical straight light projections adequate for whistlers? I would expect projection along magnetic field lines. Explain why you use vertical projection.

The propagation of lightning whistlers along the magnetic field lines is likely, and we have now modified Fig. 3 assuming the field aligned propagation for groups of rapid whistlers, combined with vertical projection for groups of JDPs. It is true that it doesn't make much difference compared to the previous version with the straight-line projection for both JDPs and rapid whistlers, as the difference is only about a few degrees in midlatitudes, where whistlers were most frequently observed. This was also the reason that we opted for a simpler version for the first submission.

Caption of Figure 3: Are those altitudes of spacecraft or of lightning? I see it in the label, but I would repeat in caption.

Done.

Line 158: Normalization is a source of potential bias is explained in the Methods part. Give a reference here.

The Methods are referenced on line 168.

Lines 159-161: For clarity, put an equation instead of verbal description.

Done.

Line 206: Clarify how cloud-to-ground may be relevant for Jupiter. I do see you mentioning this at the end of this paragraph but I would move it right to the beginning.

We moved the sentence about the relevance of CG lightning at the beginning of the paragraph on lines 233-234.

Line 211: Again, clarify why CG may be relevant

The multi-stroke CG flashes are also repetitive lightning phenomena and without knowing precisely, the mechanism of repetitive lightning signals from Jovian atmosphere we think it might be good for completeness to list the time scale of all terrestrial repetitive lightning signals, including the CG flashes.

Line 217: I would say "in addition" because it is not a very convincing argument given all the differences between Earth and Jupiter.

The sentence was removed.

Line 222: I suggest removing previous paragraph and just mention the CG pulse frequencies in one sentence

This suggestion contradicts with suggestion of reviewer #1 who requires adding discussion of the timing statistics in the list of repetitive lightning signals.

Line 245: This is a good convincing conclusion. Put it in the abstract. "Juno data suggest step-like extensions..." Separate it from assumption of the similar speeds and corresponding step lengths, which is questionable.

We changed the abstract according to the reviewer's suggestion by stressing the convincing conclusion and we moved estimation of the step lengths to the discussion. The last sentence of the abstract now reads as follows:

"The Juno data thus suggests similar step-like extensions of Jovian lightning channels after the lightning initiation, as observed during initiation processes of intracloud lightning at Earth.

Line 246: Clarify why you would expect similar velocities. This is not nearly as obvious as suggesting step-like lightning.

We clarified this assumption on lines 287-295:

In addition here are some suggestions that would make the paper easier to read for non-specialists:

There is quite a bit of data values quoted right in the paper text. It would be helpful to give a reader more systematic view on the data using tables, equations and figure details.

We have collected all the numerical values linked to the selection process of the final dataset in Table 1.

The comparison with just one data set on terrestrial lightning is interesting. However, it would be important to know how other published observations on terrestrial lightning agree with the observations presented in this paper.

We added following sentence on lines 274 -275:

"The time separations of observed pulses in examples shown in Fig. 4 correspond to the typical time scales mentioned in the list of repetitive lightning phenomena (points c and b)."

We also added some statistical results in the list describing the time scales of repetitive terrestrial lightning signals on lines 236-255. We also added relevant references.

Reported stepped structure of Jovian lightning and its timing is a robust and convincing result. However, the main conclusion of the paper is about the length of lightning steps. The assumption of similar speeds for lightning used for the steps' length calculation needs to be better justified and discussed in the paper text. It would be very interesting to discuss the range of possible lightning propagation speeds and corresponding step lengths.

We removed the statement about the lengths of the leader steps from the abstract and stressed the stepping, which was not observed before. Based on the similarities of expected composition of water and ice particles in the Jovian and terrestrial thunderclouds and due to the very similar electric breakdown fields (ref. 39, Yair et al., 1995), we do not have any good reason to place the average leader prolongation velocities outside a velocity range of 10^5 to 10^6 m/s observed on the Earth. We added the clarification of our assumptions in a new paragraph on lines 287-295.

=====

Reviewer #3

I am very impressed by this work. Electric field waveforms produced by Jovian lightning (Sup. Fig. 1), as well as the corresponding spectrograms (Fig. 1) and the associated analysis, certainly deserve to be published in Nature Communications. My comments aimed at clarification of several points are found below.

1. I think the authors could give more credit to the pioneering Galileo probe experiment. As far as I know, in that experiment, the first wideband (100 Hz to 100 kHz) field signatures of Jovian lightning were obtained below the ionosphere with an instrument (LRD) that was calibrated against terrestrial lightning (see Lanzerotti et al., 1992, 1996). Yes, their signals were dominated by the lower-frequency components, in contrast with this study, in which the signals do not appear to indicate any significant charge transfer. Maybe the reason for disparity is the ionosphere that did not allow the lower-frequency components to be recorded in the present study. In any event, this is worth discussing in the paper, particularly the lack of detectable charge transfer in the present study and the fact that LRD was fully capable of recording impulsive signals (see Fig. 11 of Lanzerotti et al., 1992), but did not see any.

References

Lanzerotti, L.J., Rinnert, K., Dehmel, G., Gliem, F.O., Krider, E.P., Uman, M.A., and Bach, J. 1996. Radio frequency signals in Jupiter's atmosphere. *Science*, 272: 858-60.

Lanzerotti, L.J., Rinnert, K., Dehmel, G., Gliem, F.O., Krider, E.P., Uman, M.A., Umlauf, G., and Bach, J. 1992. The lightning and radio emission detector (LRD) instrument. *Space Sci. Rev.*, 60: 91-109.

We modified and substantially extended the paragraph describing the Galileo probe and its measurements. The paragraph now reads as follows on lines 208-220:

“The lightning and radio emission detector onboard the Galileo probe investigated characteristics of radio frequency signals (10 Hz to 100 kHz). The capability of the instrument to detect groups of impulsive signals was proven during its calibration against terrestrial lightning, showing the characteristics of both IC and CG discharges. During the descent phase, the probe recorded radio frequency signals from a distant source, which were dominated by low frequencies with long pulses of the order of a few hundred microseconds. It was found, that the probe entered a spot with dry and stable environment, and modelling results showed that there was a very low probability that the probe could have encountered local thunderstorms. This implies that the signals received by the probe arrived from a distant storm located up to a few thousand kilometres from the descent trajectory. These signals might be compatible with our observations but the uniqueness of these measurements does not allow us to compare the results more thoroughly. Due to a long propagation below the ionosphere, the high frequency portion of the signal was probably already attenuated. “

As regards detectability of a significant charge transfer in our measurements, the high frequency part of the low frequency receiver is not able to detect signals carrying information about a large charge transfer being limited from below at 20 kHz. The low frequency part of the receiver is able to detect signals from Hz frequencies. The instrument is thus able to detect signals coming from currents flowing in lightning channels for longer time. We found difficult to discuss thoroughly the charge transfer in our previous study, as the whistler traces exhibit dispersion and their lowest frequency parts are often masked by hissy background emissions. In this study, we focus only on

impulsive signals arriving in a fast succession, which – similarly as on the Earth - are not supposed to represent larger charge transfers. We added a note on lines 302-304:

“In contrast with the sferics detected by the Galileo Probe our data set does not indicate a substantial charge transfer. This is not surprising as it is not expected during the stepping process.”

2. Lines 243-248 (see also Lines 47-49): Some clarification is needed here. First, the initial breakdown (IB) in terrestrial lightning is usually hidden inside the cloud (not imaged by high-speed cameras), and there are different theories on the nature of this process (including some kind of runaway breakdown process). In view of the relatively poor understanding of the IB on Earth, is it prudent to state that IB processes on Jupiter and on Earth are similar?

The initial breakdown process in terrestrial lightning has been investigated by many research groups using different techniques. There are numerous studies based on electromagnetic data sets acquired at different locations and seasons. It is generally accepted that the initiation process of all CG and IC lightning flashes can be recognized from groups of bipolar pulses in the radiated electromagnetic signals (Marshall et al., 2014). The IB processes have been only very rarely observed by high-speed cameras. Nevertheless, in a few documented events the luminosity increase happened simultaneously with the electromagnetic pulses (Stolzenburg et al., 2021) which confirmed the presence of currents flowing in the extending lightning channels. The discussion about the processes occurring before the first IB pulse are still under debate (Marshall et al, 2014; Kostinsky et al, 2020).

Marshall, T., W. Schulz, N. Karunarathna, S. Karunarathne, M. Stolzenburg, C. Vergeiner, and T. Warner (2014), On the percentage of lightning flashes that begin with initial breakdown pulses, *J. Geophys. Res. Atmos.*, 119, 445–460, doi:10.1002/2013JD020854.

Stolzenburg, M., Marshall, T. C., Bandara, S., Hurley, B. & Siedlecki, R. Ultra-high speed video observations of intracloud lightning flash initiation. *Meteorol. Atmos. Phys.* 133, 1177–1202 (2021).
Marshall, T., M. Stolzenburg, N. Karunarathna, and S. Karunarathne (2014), Electromagnetic activity before initial breakdown pulses of lightning, *J. Geophys. Res. Atmos.*, 119, 12,558–12,574, doi:10.1002/2014JD022155

Kostinskiy, A. Y., Marshall, T. C., & Stolzenburg, M. (2020). The mechanism of the origin and development of lightning from initiating event to initial breakdown pulses (v.2). *Journal of Geophysical Research: Atmospheres*, 125, <https://doi.org/10.1029/2020JD033191>

The previous studies (Levin et al., 1983; Yair et al., 1995) agreed, that clouds composed of water and ice, located at temperature range of between 250 and 270 K and the internal heat which drives the strong convective motion form the best conditions for separation of charges and establishing electric fields sufficient for lightning initiation. The breakdown fields calculated for Jovian water cloud located at different altitudes (6×10^5 V/m at 1 bar level to 2.3×10^6 V/m at 5 bar level; Yair et al., 1995) are very close to breakdown fields observed at 1 bar level in terrestrial thunderclouds (3×10^5 - 1×10^6 V/m). Based on the findings described above we assume that we can hypothesize that the initiation processes at Jupiter and Earth might be similar.

Levin, Z., Borucki, W. J. & Toon, O. B. Lightning generation in planetary atmospheres. *Icarus* 56, 80–115 (1983).

Yair, Y., Levin, Z. & Tzivion, S. Lightning generation in a Jovian thundercloud: Results from an axisymmetric numerical cloud model. *Icarus* 115, 421–434 (1995).

We added a new paragraph on lines 287-295 to clarify our assumptions.

Further and more importantly, I do not see how the average leader extension speed (10^5 to 10^6 m/s) and the average inter-pulse interval (about 1 ms) can be used for estimating the step length. It is usually assumed (see, for example, Carl Baum's work) that the stepped-leader channel extends only (or primarily) during the step-formation process, as opposed to inter-step intervals. The step-formation process is much faster (10^7 to 10^8 m/s) than the overall leader extension. Please clarify (maybe in the Methods section).

Even if the step formation processes on Jupiter are as fast (10^8 m/s) as on the Earth as reported by ref. 43 - Baum (2002), we are not able to confirm it using the existing Juno data. The ionospheric dispersion does not allow us to estimate the duration of rising edges of individual pulses with a sufficient time resolution. That is why we limit our estimation to an average speed of the leader prolongation happening over a group of leader steps. To clarify this, we changed "velocity" to "average velocity" and "prolongation" to "average prolongation" on lines 300 and 301 and added following sentence on lines 305-306: "Note that the velocity of the step formation might be - similarly as on Earth - faster than the average velocity of the step prolongation (Baum, 2002)." We also removed the statement about the step length from the abstract and left it for the discussion.

3. Lines 217-218: Actually, the average number of strokes per flash is 3 to 5. The quoted range from 2 to 3 is based on ALDIS data, and all LLSs underestimate this parameter. I found a recent summary of non-LLS measurements in Table 4.1 of Rakov (2016).

Reference: Rakov, V.A. 2016. Fundamentals of Lightning, Cambridge.

Corrected. We reworded the relevant text on lines 255-256 as follows: "Even though there are on average three to six strokes in terrestrial multi-stroke CG flashes and the groups of JDPs predominantly consisted of five pulses, the Jovian pulse groups were very unlikely generated by multi-stroke lightning flashes."

4. Lines 213-214: The average time interval between stepped-leader steps is usually a few tens of microseconds (see, for example, Table 7.1 of Rakov (2016)), not "units of microseconds". We corrected the average time interval between stepped leader pulses to a few tens of microseconds and added the pulse trains occurring during the propagation of dart-stepped or recoil leaders with inter-pulse intervals of units of microseconds on lines 251-254.

In summary, I recommend this paper be published after revision to address the above comments.

REVIEWER COMMENTS

Reviewer #1 (Remarks to the Author):

I am happy with how the authors have addressed my comments and I now recommend the article for publication.

Reviewer #2 (Remarks to the Author):

I recommend the paper to be published in its current form.

Thanks for detailed responses.

Ulyana

Reviewer #3 (Remarks to the Author):

The manuscript has been considerably improved by revision. I do recommend publication. My remaining concerns are relatively minor and related primarily to the modified classification of impulsive signals in lines 236-254.

Lines 237-239: The interstroke intervals for PEC and NGC strokes in [26] are 55 and 65 ms, respectively (see Table 2 in [26]). The values given by the authors, 34 and 77 ms, respectively, are taken from Table 3 of [26] and correspond to the special situation when two or more strokes went down the same channel.

Lines 248-250: The newly provided information on inter-step intervals is confusing. It appears to suggest that there is a contrast in this parameter between Florida (3.3 us) and China (17.4 us), which is not the case. The Florida value of 3.3 us taken from [34] corresponds to many simultaneously active leader branches, while the China value [35] corresponds to individual branches (also, the Chinese value is based on field, not optical, measurements). The Florida value corresponding to individual branches is 16.6 us (see Hill et al., 2011) and it is similar to the China value.

Lines 251-253: It would be better to insert “-stepped” between “dart” and “leaders”, for clarity. I recall that the study presented in [36] was extended by Rakov et al. (1996; see also their Table III).

Line 231: Replace “dart step” with “dart-stepped”.

Line 233: Consider replacing “clear ground level” with “solid surface” or something like that.

Line 302: Please explain how the step lengths were estimated (the Methods section would be an appropriate place to do so). Also, define more clearly the terms “the velocity of the step formation” (see line 305) and “the velocity of the step prolongation” (see line 306).

References

Hill, J. D., Uman, M. A., & Jordan, D. M. (2011). High-speed video observations of a lightning stepped leader. *Journal of Geophysical Research*, 116, D16117. <https://doi.org/10.1029/2011JD015818>.

Rakov, V.A., M.A. Uman, G.R. Hoffman, M.W. Masters, and M. Brook (1996) Bursts of Pulses in Lightning Electromagnetic Radiation: Observations and Implications for Lightning Test Standards, *IEEE Trans. on EMC*, 38, No. 2, 156-164.

We thank Ulyana Dyudina and Reviewer #1 for recommending our manuscript for publication. We thank Reviewer #3 for noting several inaccurate statements in the part related to comparison with terrestrial lightning and for his/her helpful comments and suggestions. We responded to all of them and revised the manuscript accordingly. (The lines are related to the manuscript with tracked changes.)

Responses to Reviewers (in blue) with Reviewers' comments in black.

=====

Reviewer #1 (Remarks to the Author):

I am happy with how the authors have addressed my comments and I now recommend the article for publication.

Reviewer #2 (Remarks to the Author):

I recommend the paper to be published in its current form.
Thanks for detailed responses.
Ulyana

Reviewer #3 (Remarks to the Author):

The manuscript has been considerably improved by revision. I do recommend publication. My remaining concerns are relatively minor and related primarily to the modified classification of impulsive signals in lines 236-254.

Lines 237-239: The interstroke intervals for PEC and NGC strokes in [26] are 55 and 65 ms, respectively (see Table 2 in [26]). The values given by the authors, 34 and 77 ms, respectively, are taken from Table 3 of [26] and correspond to the special situation when two or more strokes went down the same channel.

Corrected on lines 228 and 229.

Lines 248-250: The newly provided information on inter-step intervals is confusing. It appears to suggest that there is a contrast in this parameter between Florida (3.3 us) and China (17.4 us), which is not the case. The Florida value of 3.3 us taken from [34] corresponds to many simultaneously active leader branches, while the China value [35] corresponds to individual branches (also, the Chinese value is based on field, not optical, measurements). The Florida value corresponding to individual branches is 16.6 us (see Hill et al., 2011) and it is similar to the China value.

We changed this point as follows on lines 238-243:

“d) sequences of a stepped leader pulses of CG lightning with inter-pulse separation of a several units to a few tens of microseconds³³ (0.2 to 15.7 μ s with an average value of 3.3 μ s corresponding to many simultaneously active leader branches in Florida, US³⁴, or an average inter-step interval of 16.4 μ s identified within optical detections of individual branches in Florida, US³⁵ - newly added reference to Hill et al. 2011, and 13.9 to 23.9 μ s with a mean value of 17.4 μ s in electromagnetic measurements of individual branches in China³⁶);”

Lines 251-253: It would be better to insert “-stepped” between “dart” and “leaders”, for clarity. I recall that the study presented in [36] was extended by Rakov et al. (1996; see also their Table III).

Corrected on line 246, the reference to Rakov (1996) added on line 248.

Line 231: Replace “dart step” with “dart-stepped”.
Corrected on line 222.

Line 233: Consider replacing “clear ground level” with “solid surface” or something like that.
Replaced on line 225.

Line 302: Please explain how the step lengths were estimated (the Methods section would be an appropriate place to do so).

We have added in the Methods section (lines 343 -353):

“Even if the rapid extension of the leader in individual steps might be as fast ($\sim 10^8$ m/s) on Jupiter as on the Earth⁴⁵, we are not able to confirm this velocity using the existing Juno data. The ionospheric dispersion does not allow us to estimate the duration of rising edges of individual pulses with a sufficient time resolution. That is why we limit our estimation to an average speed of the leader prolongation happening over a group of leader steps, i.e., including the intervals between the pulses, when we assume that the leaders do not extend. We further assume that the average speed of the Jovian leader prolongation is similar to the average speed of leaders propagating in the terrestrial water clouds at a similar pressure. If we combine this average velocity of 10^5 - 10^6 m/s with the interpulse intervals resulting from our study (typically a few milliseconds, Fig. 3d), we obtain the average step length of several hundreds to a few thousand meters.”

Also, define more clearly the terms “the velocity of the step formation” (see line 305) and “the velocity of the step prolongation” (see line 306).

To make the statement clearer, we reworded the sentence on lines 298 – 299 as follows:

“Note that the velocity of individual steps might be – similarly as on Earth – much faster than the average velocity of the leader prolongation, which includes also quiet stages between the steps.”

References

Hill, J. D., Uman, M. A., & Jordan, D. M. (2011). High-speed video observations of a lightning stepped leader. *Journal of Geophysical Research*, 116, D16117. <https://doi.org/10.1029/2011JD015818>.

Rakov, V.A., M.A. Uman, G.R. Hoffman, M.W. Masters, and M. Brook (1996) Bursts of Pulses in Lightning Electromagnetic Radiation: Observations and Implications for Lightning Test Standards, *IEEE Trans. on EMC*, 38, No. 2, 156-164.

REVIEWERS' COMMENTS

Reviewer #3 (Remarks to the Author):

This is a great paper, ready to be published.